# α-Synuclein Strains: Does Amyloid Conformation Explain the Heterogeneity of Synucleinopathies?

**DOI:** 10.3390/biom11070931

**Published:** 2021-06-23

**Authors:** Simon Oliver Hoppe, Gamze Uzunoğlu, Carmen Nussbaum-Krammer

**Affiliations:** Center for Molecular Biology, Heidelberg University (ZMBH) and German Cancer Research Center (DKFZ), DKFZ-ZMBH Alliance, Im Neuenheimer Feld 282, D-69120 Heidelberg, Germany; o.hoppe@zmbh.uni-heidelberg.de (S.O.H.); g.uzunoglu@zmbh.uni-heidelberg.de (G.U.)

**Keywords:** alpha-synuclein, synucleinopathies, prions, prion-like propagation, amyloid, conformational strains

## Abstract

Synucleinopathies are a heterogeneous group of neurodegenerative diseases with amyloid deposits that contain the α-synuclein (SNCA/α-Syn) protein as a common hallmark. It is astonishing that aggregates of a single protein are able to give rise to a whole range of different disease manifestations. The prion strain hypothesis offers a possible explanation for this conundrum. According to this hypothesis, a single protein sequence is able to misfold into distinct amyloid structures that can cause different pathologies. In fact, a growing body of evidence suggests that conformationally distinct α-Syn assemblies might be the causative agents behind different synucleinopathies. In this review, we provide an overview of research on the strain hypothesis as it applies to synucleinopathies and discuss the potential implications for diagnostic and therapeutic purposes.

## 1. Introduction

The accumulation of misfolded α-synuclein (SNCA or α-Syn) is implicated in several neurodegenerative diseases known as synucleinopathies, which include Parkinson’s disease (PD), dementia with Lewy bodies (DLB), and multiple-system atrophy (MSA) [1,2,3]. Human α-Syn is a small, 140 amino acid protein that is encoded by the *SNCA* gene on chromosome 4 [4,5]. Under physiological conditions, native α-Syn is in equilibrium between a cytoplasmic intrinsically disordered state and a membrane-bound α-helical state [6]. α-Syn is predominantly expressed in the central nervous system (CNS) and peripheral nerves, where it is highly concentrated at the pre-synaptic terminals. Its specific cellular function has not been fully elucidated, but it has regulatory activities in synaptic plasticity and vesicle trafficking [7]. Under pathological conditions, α-Syn assembles into well-ordered, β-sheet-rich amyloid fibrils that accumulate into large intracellular inclusions [6].

The causal role of α-Syn in disease pathogenesis is supported by genetic evidence. Several missense mutations in the human *SNCA* gene that lead to destabilizing single amino acid changes in the protein (A53T, A30P, E46K, H50Q, G51D, and A53V) are linked to autosomal-dominant inherited early-onset PD [8,9,10,11,12,13]. Furthermore, A18T and A29S amino acid substitutions have been recently described in late-onset sporadic PD patients [14]. In addition, multiplications (duplication or triplication) of the *SNCA* gene locus have also been associated with early-onset familial parkinsonism [13,15,16,17,18]. An elevated *SNCA* gene dosage correlates with increased disease penetrance, earlier disease onset, and more severe PD symptoms [19,20]. Additional evidence, albeit less prominent, points to involvement of α-Syn levels and mutations in the etiology of DLB and MSA. For instance, A53T, E46K, and E83Q missense mutations and *SNCA* multiplications have been observed in families with mixed PD and DLB phenotypes [21,22], while G51D and A53E mutations have been found in cases with mixed PD/MSA pathogenesis [23,24]. In addition, genetic variations in *SNCA* affecting α-Syn levels rather than amino acid sequences have been identified in MSA patients [25,26,27,28]. Nevertheless, these familial forms represent a minority (less than 10%), while the majority of synucleinopathy cases are considered idiopathic (or sporadic), meaning that the disease occurs spontaneously with no known cause [20,22,25].

Apart from being historically classified as movement disorders, synucleinopathies are multifactorial diseases with motor symptoms and non-motor symptoms that display great heterogeneity in their pathology and clinical manifestation. In 1817, James Parkinson described the clinical symptoms of PD as resting tremor and movement abnormalities (bradykinesia and akinesia) [29]. Almost a century later, in the early 1900s, post-mortem examination of PD patients led to the discovery of the typical cytoplasmic inclusions in the neuronal soma or axon and dendrites, known today as Lewy bodies and Lewy neurites, respectively. The presence of Lewy pathology became the histopathological hallmark of PD and DLB [30,31,32,33]. In the late 1990s, when *SNCA* mutations were linked to familial forms of PD, α-Syn was shown to be a major protein constituent of Lewy bodies and Lewy neurites [1,2,9]. In contrast, the histological hallmark of MSA is not Lewy pathology but the formation of α-Syn-positive cytoplasmic inclusions in oligodendrocytes, termed glial cytoplasmic inclusions (GCIs) or Papp–Lantos bodies [26,34,35].

Although α-Syn is the major protein component of pathological inclusions, along with membranous organelles [36,37], they differ not only in their subcellular localization, but also in the brain regions affected in the individual diseases. In the case of MSA, α-Syn-positive inclusions can be found throughout the brain, with the highest density of GCIs found in the basal ganglia, while some neuronal cytoplasmic and nuclear inclusions are found in many additional regions, including the putamen, substantia nigra, and amygdala [38]. In PD, Lewy pathology predominantly develops in the nigrostriatal area and at the brainstem concurrent with severe loss of dopamine neurons in the substantia nigra [39,40]. In DLB, Lewy pathology affects mainly the neocortex and the limbic system, with less severe dopaminergic cell loss in comparison to PD [40,41,42]. The two Lewy body diseases further differ at the clinical level, as DLB causes cognitive impairment in the form of fluctuating states of alertness and consciousness, visual hallucinations, in addition to spontaneous parkinsonism-like motor symptoms [40,42]. Moreover, non-motor symptoms such as a decline in the olfactory sense, rapid eye movement sleep disturbances, and gastrointestinal dysfunction occur much earlier than motor symptoms in PD patients, unlike in DLB, where there is only a one year delay before motor symptoms begin [40,41,42]. Clinically, MSA is characterized by varying degrees of autonomic failure, cerebellar ataxia, and motor dysfunction that resemble PD, making its early diagnosis challenging [26,38,43]. MSA is further divided into two subtypes depending on the type of symptoms present at the time of diagnosis: the parkinsonian subtype (MSA-P), which is characterized by parkinsonism, and the cerebellar subtype (MSA-C), where cerebellar ataxia is prominent [44]. Albeit very rare, MSA is the most aggressive of the synucleinopathies, with the earliest mean age of onset (54 years), as well as a faster progression compared to PD and DLB, leaving the patients disabled within a short period of time [26,40,44].

This description clearly highlights that the misfolding and accumulation of a single protein can lead to a variety of clinical and pathological manifestations, a phenomenon that is not fully understood [28,42,45]. Recent studies have established that α-Syn is capable of adopting distinct fibrillar conformations in vitro, which can induce different pathological phenotypes when injected into rodents [46,47,48]. This has led to the hypothesis that distinct α-Syn pathologies may be caused by structurally distinct amyloid assemblies of α-Syn [49]. In this review, we first introduce the concept of conformational strains, originally established in the context of prion diseases, and then present the current literature on conformational α-Syn variants. In particular, we will focus on recent structural analyses of patient-derived α-Syn aggregates. Finally, we discuss the potential clinical implications of the current findings regarding α-Syn strains.

## 2. Prion Strains

### 2.1. What is a Prion?

As is the case for α-Syn and its associated diseases, the synucleinopathies, the prion protein (PrP) is implicated in a wide range of clinically recognized diseases, collectively called the transmissible spongiform encephalopathies (TSEs). TSEs are inevitably fatal neurodegenerative diseases that occur within the mammalian class of life, across three different phylogenetic orders: as transmissible mink encephalopathy (TME) and feline spongiform encephalopathy (FSE) in carnivores; as scrapie, chronic wasting disease (CWD), and bovine spongiform encephalopathy (BSE) in even-toed ungulates (sheep, deer, and cattle, respectively); and as Creutzfeld–Jakob disease (CJD), Gerstmann–Sträussler–Scheinker disease (GSS), fatal familial insomnia (FFI), and kuru in humans [50].

In TSEs, the aggregation of a single protein is considered a central unifying hallmark of pathology across the disease family. The endogenous prion protein (PrP^C^, for cellular PrP), a glycosylphosphatidylinositol-anchored cell surface protein that is highly abundant in the peripheral and central nervous systems (PNS and CNS, respectively), undergoes conformational changes from its native α-helical structure to its pathogenic, β-sheet rich conformation (PrP^Sc^, for PrP scrapie) [51,52,53]. Conformational conversion of PrP^C^ into this misfolded state can occur sporadically or through an existing PrP^Sc^ seed, acting as a template. In this way, PrP^Sc^ amyloid aggregates are thought to grow on either end by monomer addition [54,55]. Larger polymers of aggregated PrP^Sc^ may then fragment into smaller seeds, causing an amplification of seeded misfolding of PrP^C^ as new interfaces for monomer conversion are generated [56,57,58], which themselves may template misfolding as they spread throughout CNS structures.

The majority of human TSE cases have either a sporadic (e.g., sCJD, 85–90% of cases) or genetic (gCJD, 5–15% of cases) etiology; however, as the word prion (derived from “proteinaceous infectious particle”) already implies, a unique feature of prion diseases is that they can be transmitted within a species (<1% of cases) and even between species in certain circumstances. This transmissibility has led to the initial assumption that the diseases must have a bacterial or viral cause [59]. The unanticipated discovery that the disease may have an acquired origin through environmental exposure to infectious PrP^Sc^ seeds subsequently led to the “protein only hypothesis”, which postulates that the misfolded PrP^Sc^ prion is the sole disease-causing agent.

### 2.2. What is a Prion Strain?

From a clinical perspective, all human prion diseases progress to a similar endpoint—akinetic mutism (no movement, no speech) and subsequent death within a disease course of one year or less. This progression is relatively rapid in comparison to other neurodegenerative diseases such as Alzheimer’s disease (AD) or PD. The clinically observable symptoms in the prodromal phase of the disease before the onset of unambiguous symptoms, however, can vary widely within the same disease [60].

For instance, “classical presentation” of sporadic CJD (sCJD) consists of global cognitive decline (dementia) in combination with motor symptoms manifesting as coordination difficulties (cerebellar ataxia); involuntary, irregular twitching (myoclonus); and other symptoms such as stiffness, tremor, and difficulties with fine motor skills (pyramidal and extra pyramidal symptoms) [61]. Different clinical manifestations are related to the different areas of the brain affected by misfolded prions; however, a wide range of atypical disease presentations are known, in which symptoms manifest purely cognitively or purely through motor symptoms. In some cases, the first clinical signs of CJD are psychiatric, behavioral in nature, and are misdiagnosed as psychiatric depression or psychosis [50]. As symptoms are so varied and a diagnosis of CJD carries such weight for patients and families (along with genetic implications for relatives), a final diagnosis of CJD is often only reached postmortem, when histopathological profiles of lesions and the presence and distribution pattern of PrP^Sc^ inclusions can be analyzed. How can this clinical heterogeneity of symptoms be explained for prion diseases, where the misfolded prion protein is the single disease-causing agent according to the protein-only hypothesis?

Interestingly, variant CJD (vCJD), which arises from consumption of meat from BSE-affected cattle, has a relatively stereotypical disease phenotype, in stark contrast to sCJD described above [62,63]. Patients initially develop psychiatric and behavioral symptoms, along with cerebellar ataxia, often accompanied by myoclonus. Cognitive decline commonly develops later in the disease course. In addition, brains of vCJD patients show consistent neuropathological patterns at autopsy. Spongiform change, accompanied by neuronal loss and astrogliosis, can be seen mainly in the basal ganglia and thalamus of the cerebrum, but also spreads throughout the cerebral cortex, while PrP amyloid plaques can be detected in the cerebrum and cerebellum [63].

An explanation for the diversity of disease phenotypes seen in sCJD on the one hand and the relative consistency of clinical symptoms and neuropathology observed in vCJD on the other could stem from the prion strain hypothesis [50,64]. The hypothesis states that aggregated PrP^Sc^ can adopt distinct amyloid conformations causing distinct disease phenotypes. If these phenotypes remain consistent upon serial transmission of each prion variant between hosts of the same and sometimes even different species [65,66], they are referred to as different prion strains. Following this hypothesis, vCJD would be caused by a single PrP^Sc^ strain, whereas the different sCJD disease manifestations would be caused by a number of other structurally distinct prion strains (Figure 1).

The experimental basis for the prion strain hypothesis comes from observations made in models of animal prion diseases. The serial transmission of infectious PrP^Sc^ inoculum in model animals, such as hamster and mouse, has led to the emergence of many distinct prion strains each with consistent biochemical and clinical characteristics. For example, the infection and serial passage of PrP^Sc^ material from TME-positive mink in Syrian hamsters has led to the isolation of two distinct prion strains named drowsy (DY) and hyper (HY) [66,67]. Prion strains such as DY and HY, observed in animal models, can be differentiated by their in vivo and in vitro properties. Important in vivo characteristics of prion strains include the incubation period (meaning the time elapsed between experimental inoculation with PrP^Sc^ material and the onset of clinical symptoms), the histological profile of vacuolization, and the presence of PrP^Sc^ inclusions found in the postmortem brain. In addition, the clinically observable disease phenotype is considered (hamsters behaving “drowsy” or “hyper” for example). In vitro characteristics of prion strains, which are highly consistent and strain specific, are based on certain biochemical assays. These include the overall resistance of the isolated material to proteinase K (PK) digestion or denaturation via chaotropic salts, as well as the glycosylation profile of PrP^Sc^ [65,68,69]. The sum of in vivo and in vitro characteristics that a certain well defined prion inoculum has upon serial passage in a certain host allows the definition or discrimination of a prion strain.

The variable sensitivity of prion strains to PK digestion and denaturing conditions indirectly suggests that the differences observed among strains are conformational in nature; however, rigorous experiments with different PrP^Sc^ aggregates, which could address this question directly, are hampered by their infection potential and associated stringent biological safety guidelines. Fundamental aspects of prion biology have, therefore, been addressed extensively using yeast prion proteins, such as Sup35, which cannot harm humans. Sup35 is a translation termination factor in yeast. Due to a glutamine–asparagine-rich (prion) domain, it can acquire a non-native amyloid conformation and aggregate; thereby, more of the native termination factor is sequestered away from its mRNA target sequence, eventually resulting in a partial loss-of-function phenotype, [PSI+], in which stop codons are ignored [70]. This phenotype is based on the amyloid aggregation of Sup35, along with its seeded conversion, fibril growth, and fragmentation, and clearly shows a non-genetic distribution into yeast daughter cells. The prion biology of Sup35 has been explored in great detail [71]. Amyloid growth could be shown to occur by templated monomer addition on fibril ends [72]. Several Sup35 conformational variants causing phenotypically different [PSI+] variants have been described [73]. Remarkably, it could also be demonstrated that two distinct Sup35 conformational variants, Sc4 and Sc37, can arise spontaneously in vitro from the same pool of native monomer, simply through incubation at different temperatures (4 and 37 °C, respectively). Sc4 and Sc37 produce strong and weak [PSI+] phenotypes, respectively, which are stably inherited to the progeny [74]. The observed phenotypic differences were attributed to conformational differences between Sc4 and Sc37 through structural studies based on the combination of nuclear magnetic resonance (NMR) and hydrogen–deuterium (H/D) exchange [75]. While both strains have overlapping amyloid cores, encompassing the majority of the Gln/Asn-rich ‘steric zipper’ prion domain, in Sc37 this amyloid core is extended to contain the first 70 amino acids of the Sup35, neatly explaining the greater fibril stability observed in Sc37 fibrils [75]. The detailed experiments with Sup35 and other prions in yeast have shed light on the fundamental mechanisms of prion biology, which are likely of general relevance for prions and prion-like proteins. They have also demonstrated that key characteristics determining strain phenotypes, such as the fibril growth rate and frangibility, can be mediated entirely through different amyloid conformations arising from the same pool of monomers [75].

While the evidence supporting the prion strain hypothesis from basic yeast models and mammalian animal models is extensive, it is still largely indirect. High-resolution structural data for the materials found in patients with clinically different prion diseases would be necessary to pinpoint whether different amyloid conformations of PrP^Sc^ indeed correlate with different clinical manifestations seen in sCJD and vCJD; however, studies of this nature in the prion field have always been complicated by more rigorous biological safety standards surrounding infectious PrP^Sc^ material. As such, the recently published first high-resolution cryogenic electron microscopy (cryo-EM) study revealing the structure of infectious full-length PrP^Sc^ material originating from scrapie sheep and passaged in Syrian hamsters represents an important milestone [76]. The analyzed prions consisted of amyloid fibrils, in which residues 95–227 comprised the stacked fibril core, forming a parallel in-register β-sheet, the so called “Greek key” architecture [77]. How this structure compares to material in human patients and whether or not conformationally distinct strains can be identified in CJD patients will have to be investigated in similar studies in the future.

## 3. α-Syn Strains in Synucleinopathies

### 3.1. α-Syn Has Prion-Like Properties and Can Adopt Distinct Conformations In Vitro

The idea of a specific amyloid conformation or strain being at the center of a highly diversified pathogenesis caused by a single or few proteins is not limited to the prion field. Indeed, there is growing clinical and scientific evidence that α-Syn shares many properties with prion proteins [78,79,80]. The first indication for prion-like propagation of α-Syn came from post-mortem examination of PD patient brains. In 2003, Braak and co-workers described a staging system for PD based on the gradual appearance of α-Syn inclusions along a characteristic spatiotemporal trajectory (Figure 2). This finding led them to speculate that a hypothetical unknown pathogen invading the nervous system could be the reason behind idiopathic PD [81,82]. A few years later, in 2008, two independent post-mortem studies on PD patients who had received embryonic neuronal transplants reported that the grafted neurons had Lewy-body-like inclusions. From these observations, it was inferred that pathological α-Syn had spread from the host to the grafted tissue in a prion-like manner [83,84]. This behavior of α-Syn aggregates has been recapitulated in animal models of PD [85,86,87,88,89], DLB [90], and MSA [91,92]. Moreover, experiments in tissue culture cells demonstrated the release and uptake of α-Syn across different cell types [93,94,95]. The prion-like propagation hypothesis for α-Syn was further supported by studies showing that recombinant fibrillar α-Syn species can seed the aggregation of soluble α-Syn and propagate their specific conformation in vitro [80,96,97], as well as trigger the conversion of endogenously expressed soluble α-Syn into self-propagating aggregates in cell culture and animal models [85,86,87,93,95,98,99,100]. These studies further demonstrated that α-Syn species can be transmitted not only between neurons [93,94] and from neurons to oligodendrocytes [91,92] in the brain, but also from distinct tissues in the periphery such as the gut [88] and muscle [89] to the brain.

One of the very first indications of the existence of α-Syn conformational strains came from a study by Ronald Melki and colleagues, which showed that wild-type α-Syn can assemble into two distinct polymorphs in vitro under defined buffer compositions with different salt concentrations [46]. These two high-molecular weight assemblies of α-Syn, named “fibrils” and “ribbons”, exhibit clear structural differences as visualized by electron microscopy (EM) [46]. Fibrils and ribbons further differ in their thermal stability and resistance to proteolytic digestion by PK. The structural properties of fibrils and ribbons obtained via NMR spectroscopy revealed that both polymorphs are composed of β-sheet-rich subunits; however, the packing of individual α-Syn molecules within these assemblies is different [46]. Finally, the polymorphs vary in their seeding capacity and toxicity when applied to cultured cells [46]. Further early evidence for the existence of α-Syn strains was reported by Virginia Lee and colleagues, showing that two distinct synthetic α-Syn fibrils differ in their ability to cross-seed Tau aggregation and induce toxicity in cultured neurons [101]. These two types of α-Syn fibrils can be distinguished by their PK digestion patterns and by using conformation-specific antibodies, indicative of their structural differences [101]. This intriguing discovery that recombinant α-Syn conformational strains differentially affect cultured cells led to the question of whether this might also hold true in vivo in animal models. Indeed, subsequent studies revealed that in-vitro-generated α-Syn polymorphs were able to affect different brain regions and induce distinct disease phenotypes in rodents that were preserved upon serial passaging [47,48,102].

Together, these studies demonstrated that monomeric α-Syn has the intrinsic potential to assemble into distinct fibrillar conformations under different physicochemical conditions in vitro, with strain specific structural and biochemical features, and that these strains can propagate their signature structure in a prion-like fashion and cause a distinct pathology in cultured cells and animal models.

### 3.2. Are There Different α-Syn Strains Behind Synucleinopathies?

It is tempting to assume that the clinical and pathological variability in synucleinopathies might be explained by differences in α-Syn aggregate structures, analogous to prion diseases, given that the biology of α-Syn resembles the prion biology in many ways. While the experiments with recombinant α-Syn conformers are invaluable for the proof of principle that a certain fibril structure can cause a particular pathological signature, they do not allow the reverse conclusion that the different synucleinopathies are actually caused by different α-Syn strains. The biggest advantage of using purely synthetic α-Syn fibrils is that they have been generated under well-defined experimental conditions. This provides high reproducibility of specific fibril types, allowing researchers to focus on studying a precisely defined and well-characterized conformation; however, it is unclear whether the fibrils generated in a test tube fully represent the structures of amyloid α-Syn in cellular deposits in patients and whether they are indeed structurally distinct in different synucleinopathies. For example, “ribbons” are generated under non-physiological salt concentrations, making it questionable whether this structure can be formed and replicated in vivo [46]; therefore, it is essential to isolate and substantially characterize α-Syn species that are formed in the crowded cellular environment under the influence of interacting factors, such as the protein quality control machinery, lipid membranes, and organelles, in order to shed light on the relationship of α-Syn conformations with disease phenotypes of the various synucleinopathies.

Studies with α-Syn isolated from synucleinopathy patients indeed showed strong differences in propagation and seeding capacity [103,104,105,106,107]. MSA-patient-derived material robustly seeded the aggregation of soluble α-Syn in human embryonic kidney (HEK) cells expressing an A53T α-Syn reporter construct [103] and caused CNS dysfunction in TgM83 mice that overexpress human A53T α-Syn [104,105], while PD-patient-derived material failed to seed α-Syn in HEK cells [103] or to cause neurodegeneration in these mice [105]. Using a related cell culture model, Yamasaki et al. observed a similar trend with higher seeding activity for MSA-patient-derived α-Syn in comparison to material isolated from PD patients [106]. MSA- and PD-patient-derived samples further induced distinct α-Syn inclusion morphologies [106]. This was also in line with another study, which reported better seeding activity in mice for samples from MSA patients than those obtained from patients with Lewy pathology [107]. Differences in seeding capacity do not only exist between individual synucleinopathies. High heterogeneity in fibril templating activity has also been reported in CSF and brain extracts exclusively from DLB patients, which may be a contributing factor to the variable disease phenotypes [108].

Variability in biochemical properties, such as solubility as an indirect indication of the presence of different amyloid structures in different synucleinopathies were reported as early as 2001 [109] and repeatedly observed [103,104,105,106,107]; however, correlation of these features with structural polymorphs has been hampered by the small amounts of α-Syn aggregates that can typically be recovered from patient samples.

The recent development of in vitro cell-free prion-like amplification techniques has enabled the scaling-up of material for more detailed structural studies [110,111,112,113]. Using real-time quaking-induced conversion (RT-QuIC), α-Syn was amplified from PD and DLB patient samples and analyzed via Raman spectroscopy, transmission electron microscopy (TEM), and atomic force microscopy [110]. These experiments showed that seeding conversion activity was dependent on the presence of β-sheet-containing rod-shaped fibrils in DLB-patient-derived samples, which were absent in PD-patient-derived samples [110]. In another study, α-Syn aggregates were amplified from individuals with PD, MSA, and DLB by protein misfolding cyclic amplification (PMCA), a similar technique to RT-QuIC that also employs the seeded propagation principle to enrich misfolded proteins in vitro [111]. Side-by-side comparison of α-Syn derived from the three synucleinopathies and recombinant polymorphs revealed morphological differences in TEM analysis, in addition to the differences in their seeding characteristics and toxicity phenotype in human cells and rats [111]. PD- and MSA-patient-derived α-Syn assemblies have a flat and twisted shape, similar to the recombinantly generated “ribbons”, while α-Syn species amplified from DLB patients are cylindrical, similar to the recombinant polymorph “fibrils” [111]. Shahnawaz et al. performed cryogenic electron tomography analysis on aggregated α-Syn amplified from PD and MSA patient samples using PMCA [113]. The three-dimensional tomograms revealed that both PD- and MSA-patient-seeded fibrils are composed of two protofilaments that intertwine to form a left-handed helix; however, the periodicity of fibril twists varies greatly between PD and MSA [113]. α-Syn fibrils from PD patients are made up of long stretches of straight filaments with less frequent helical twists ranging from 76.6 to 199 nm length, whereas α-Syn filaments from patients with MSA contain more frequent, shorter twists varying from 46 to 105 nm in length [113].

Although these findings strongly support the hypothesis that α-Syn fibrils are composed of different α-Syn conformers in different pathological manifestations of α-Syn, higher structural detail with sub-nanometer atomic resolution is obligatory to provide definitive evidence. Molecular-level insights into the structure of α-Syn aggregates were first obtained from various recombinant α-Syn fibrils by cryo-EM [114,115] and NMR spectroscopy [77]; however, the structure of amyloid α-Syn derived from human brains had not been characterized at the molecular level until recently. Cryo-EM analysis of pathological α-Syn isolated from brains of MSA and DLB patients showed that α-Syn fibrils in MSA patients consist of two types of twisted filaments that co-exist at different ratios in individual MSA patients, while fibrils from DLB patients lack the twisted structure and are thinner [116]. Type I and type II MSA α-Syn filaments have extended N-terminal arms and a compact C-terminal body and are each made up of two non-identical protofilaments that asymmetrically pack against one another [116]. The core of the larger protofilament of type I filaments comprises amino acid residues G14–F94, while the smaller protofilament core consists of residues K21–Q99. In type II filaments, the fibril cores are composed of amino acid residues G14–F94 and G36–Q99 [116]. Intriguingly, the protofilaments form a cavity encompassing an additional density of unknown origin, which indicates the presence of a non-proteinaceous yet unidentified molecule [116].

Of note, Lövestam et al. reported that PMCA does not fully replicate the fibril structure present in patient brains used as seeds in the in vitro amplification assay [117]. This implies that the amyloid amplification method used in the respective studies may have influenced the examined structures. This could explain why some studies observed similarities between the structures of patient-derived and recombinant α-Syn polymorphs [111], while others reported significant differences [112,116]. The presence of additional cellular components could have an influence on the amyloid conformation obtained in vivo, as indicated by the aforementioned structural data for extracted MSA filaments analyzed directly without an in vitro amplification step [116,117]. Another important point regarding in vitro amplification methods is that the standard procedure involves the use of thioflavin T (ThT) for fibril detection; however, α-Syn is capable of forming ThT-invisible polymorphs that escape ThT-based monitoring, which could lead to misinterpretations regarding the structure and amount of fibrillar α-Syn in patient-derived samples [113,118]. Despite such caveats, these studies collectively provided direct evidence for the existence of the structural heterogeneity of α-Syn in synucleinopathy patients, supporting the hypothesis that different conformational strains of α-Syn may underlie the distinct pathology of these diseases (Figure 3).

Similar observations regarding prion-like propagation and spreading of pathology and the existence of conformational strains have also been extended to other proteins, most notably Tau and the associated tauopathies. A comprehensive review of these findings was recently compiled by Vaquer-Alicea et al. [119].

### 3.3. How Could Different α-Syn Strains Cause Distinct Phenotypes?

The most obvious explanation for the different disease manifestations of synucleinopathies is that different regions and cell types of the brain are affected by aggregated α-Syn [39]. This selectively damages the function of the respective cells, which could explain the different symptoms. What causes the occurrence of different pathological lesions and distinct distribution patterns? α-Syn is predicted to have co- and post-translational interactions, with over 100 proteins involved in various cellular processes including transcription, translation, folding, trafficking, secretion, and degradation [120]. Given this broad range of potential interactions, one could speculate that the various fibrillar conformers of α-Syn interact differently with specific cellular components of certain cell types, consequently leading to different pathological and clinical phenotypes [39].

A possible molecular mechanism that may underlie the heterogeneity of synucleinopathies was proposed by Ihse et al., showing that there are cell-type-specific differences in α-Syn internalization [121]. While the uptake of *α*-Syn fibrils in neurons and oligodendrocytes is dependent on cell surface heparan sulfate, it is dispensable for uptake in astrocytes and microglia. Interestingly, aggregate conformation also affects α-Syn internalization, with ThT-positive amyloid fibrils being more efficiently taken-up compared to ThT-negative oligomeric species [121]. Distinct fibrillar *α*-Syn polymorphs also vary in their extent of binding and accumulation in the neuronal plasma membrane [122]. Moreover, Suzuki et al. found that two α-Syn conformers differed in their ability to inhibit the activity of the 26S proteasome [123]. Conformational differences in α-Syn fibrils affect their cellular clearance, leading to different degrees of toxic phenotypes [123]. These studies provide evidence that the conformation does indeed impact the interactions of aggregated α-Syn with other cellular components, ultimately resulting in distinctive phenotypes. Whether this phenomenon also applies for α-Syn strains in patient brains remains to be elucidated.

The finding that the fibril structure can influence how different cell types and cellular constituents are affected by α-Syn raises the question of whether the reverse is also true. Could the cell type and the diversity of its cellular components be the reason behind the formation of conformational variants in the first place? Notably, pathological α-Syn isolated from the brains of patients with glial cytoplasmic inclusions (GCIs) or Lewy pathology did not display any cell-type-specific seeding preferences in vitro, albeit having distinct cell-type-specific distributions in diseased brains [107]. In contrast, however, the cell type played a critical role in determining which type of α-Syn inclusions were formed, with only oligodendrocytes but not neurons giving rise to an α-Syn variant with GCI-like properties [107]; hence, the authors concluded that the cellular environment governs the emergence of a particular α-Syn variant [107]. In this context, cellular kinases regulating α-Syn phosphorylation status may play a role in the formation of different α-Syn strains. Recombinant α-Syn phosphorylated at serine-129 (pS129 α-Syn), a common post-translational modification associated with synucleinopathies, indeed forms a distinct conformer with increased cytotoxicity compared to fibrils composed of unphosphorylated α-Syn [124]. α-Syn fibril structure can be influenced not only by intrinsic cellular factors, but also by external pathological agents. Exposure to lipopolysaccharides, bacterial endotoxins, can also induce the formation of a distinct α-Syn conformational strain in vitro, with distinct pathological properties in vivo [125].

Overall, these studies suggest that interactions between genetic and environmental factors, including infectious agents, lead to the formation of distinct α-Syn strains and influence their prion-like behavior and pathology; likewise, conformation affects the interactions of α-Syn amyloids with cellular components. Future research will reveal whether and which factors may determine a person’s susceptibility to a certain α-Syn strain and synucleinopathy.

### 3.4. Can Synucleinopathies Be Infectious?

An examination of the strain hypothesis is underway in all neurodegenerative disease fields that have been linked to amyloid protein aggregation and propagation [126]. Given the large overlap of clinical symptoms and the biochemical and biophysical properties of prion and prion-like proteins, what differentiates the two?

What principally separates prion diseases from other protein misfolding diseases is that prion diseases are by definition infectious [127,128]. In most cases, transmission is restricted to individuals within a species, but on rare and isolated occasions prions can cross species barriers [127,128]. While TSEs are considered transmissible diseases in humans, it is important to note that this is extremely rare. For CJD, recognized paths of infection include the use of amyloid-contaminated surgical instruments, injection of contaminated pituitary-gland-derived growth hormone from human cadavers, as well as grafts of human tissues such as dura mater or the cornea. Such variants of CJD arising from accidental inoculation during medical procedures are collectively termed iatrogenic (iCJD). Other rare instances of human-to-human TSE transmission are documented for cases of kuru, where ritualistic endo-cannibalism of CNS tissue results in PrP^Sc^ infection. While transmission of prion diseases between humans has been only observed in isolated cases and special circumstances, this is clearly not the case for certain forms of animal TSEs. Chronic wasting disease (CWD) and scrapie are known to spread rapidly through populations, and infectious PrP^Sc^ particles are readily detectable in the urine, feces, and saliva of afflicted animals [129]. Lastly, the most widely acknowledged incident of inter-species prion infectivity occurred in the 1980s and 1990s, when consumption of meat from BSE-stricken cattle caused a new variant of CJD (vCJD) in humans [127]. This last incidence of prion transmission from animal to human in particular fueled global concerns towards TSEs, resulting in import bans on British beef and the precautionary slaughter of 4.4 million cattle in the UK [130].

Despite the overwhelming resemblance of α-Syn propagation to prion propagation, synucleinopathies are not considered infectious and there is no direct evidence for inter-individual transfer of synucleinopathies [79]. Epidemics such as vCJD or kuru, arising from the spread of infectious amyloid seeds between individuals, have never been documented for α-Syn. Epidemiological data, indicating an iatrogenic spread of synucleinopathies resulting from surgical procedures or tissue grafts, have not emerged to date. Nevertheless, α-Syn can clearly form amyloid-type aggregates that template misfolding of native monomers and injected amyloid fibrils of α-Syn are able to seed de novo misfolding throughout interconnected CNS structures in model animals [86,87,98]. Moreover, aggregated α-Syn materials from MSA, PD, and DLB patients have been shown to cause neuropathology and neurodegeneration reminiscent of synucleinopathies in mammalian models after intracerebral injection. Notably, injection of aggregated α-Syn from PD and DLB patient brains causes accumulation of pathogenic α-Syn aggregates in PD-relevant brain regions such as the substantia nigra pars compacta, as well as early signs of neurodegeneration in wild-type non-human primates (macaques) and mice, even in the absence of a human *SNCA* transgene [131]. A transmission of synucleinopathies between individuals of the same or different species, thus, seems to be possible in principle, albeit requiring the targeted injection of concentrated disease material purified from the CNS for example. Whether or not there are particular strains of amyloid α-Syn, which could be truly infectious in certain circumstances, i.e., via surgical inoculation or consumption of contaminated CNS material, cannot be excluded entirely. This issue will require further monitoring in the future, especially with regard to the infection potential of reusable surgical equipment, from which highly stable amyloid-type aggregates may only be removed using specialized procedures that exceed current sterilization standards [132].

As fundamental aspects of prion biology seem to be shared both by PrP and α-Syn, the question arises why the associated diseases, TSEs and synucleinopathies, have such drastically different clinical manifestations. Why are TSEs so much more aggressive, usually leading to akinetic mutism and subsequent death within a year of the onset of symptoms, while synucleinopathies have a relatively slow progression? And why are only prion diseases infectious? Biochemical and high-resolution imaging techniques show that both pathological PrP^Sc^ and α-Syn materials contain amyloid fibrils; however, their individual size, growth kinetics, frangibility, and environmental stability could differ, affecting their propagation capacity and ultimately determining disease dynamics and infectivity [133]. Greater accessibility of a given fibril structure to the cellular quality control system (e.g., chaperone networks) could lead to higher fibril fragmentation [134,135]. This could explain the pathological differences between the two disease classes, as small aggregates or oligomers in related amyloid proteins have been associated with accelerated spreading and higher toxicity [127]. The individual fibril architectures could further influence the interactions with particular essential proteins and contribute to different degrees of toxicity in distinct brain areas. In addition, other factors such as cellular localization (at the outer cell membrane in the case of PrP^C^ vs. cytosol in the case of α-Syn), expression levels, and cellular functions are also likely to play a role.

Whether or not the classification into prion and prion-like or prionoid proteins is useful in light of advances in the field over the past decade is a topic that is currently under debate [136].

## 4. α-Syn Strains in Diagnostic and Clinical Applications

Data from high resolution structural studies and experiments in mammalian models of synucleinopathies, as reviewed above, seem to be in agreement with the prion strain hypothesis, stating that amyloidogenic proteins such as α-Syn, Tau, and PrP acquire a range of different fibril conformations, which correlate with distinct clinical phenotypes. Although additional evidence for the hypothesis is necessary before the idea can be regarded as substantiated, it is worthwhile thinking about the therapeutic implications this would have for patients. Can in-depth knowledge of different strain conformations associated with these diseases be leveraged into beneficial clinical applications and therapeutic strategies? Would this be an avenue for personalized medicine?

Targeting amyloid aggregates with discrete conformations directly, e.g., for cellular degradation by the 26S proteasome or autophagic pathways, may not be a promising course of action. As has been demonstrated in the prion field, targeting certain strain conformations specifically with small-molecule inhibitory compounds is only effective for a short while, before adaptation and resistance towards the compounds in question occurs. It seems that prion strains can “mutate” around the presence of inhibitory compounds targeting certain conformations [137,138,139]. An explanation for this phenomenon comes from the so-called “conformational selection” model, posited by John Collinge and colleagues, to interpret strain mutation and transmission characteristics observed in TSEs [140]. In this model, prion strains are composed of a “cloud” of similar but subtly distinct PrP^Sc^ conformations rather than a single structure, typically with a few major and several minor conformational variants (Figure 4), the exact composition of which is determined by host (genotype, tissue of origin, expression level of PrP^C^, etc.) and proteopathic seed (genotype, sequence, and conformation of the seed, etc.)-specific parameters. Since amyloidogenic proteins share many fundamental features (discussed above), it is tempting to extend this hypothesis beyond bona fide prions [141] to α-Syn strains in synucleinopathies, although there is no experimental evidence for this to date. Following this concept, eliminating one or several conformational variants of a certain strain would only facilitate the growth and spreading of a different conformation within the cloud, making it an inappropriate point of attack for clinical interventions.

A more promising route of therapeutic intervention based on strain identity seems to be to targeting the available pool of native monomers and preventing them from being converted into a toxic self-propagating fold. This could conceivably be achieved on the one hand by reducing the levels of monomeric α-Syn, for instance via RNAi or CRISPRi, targeting the *SNCA* mRNA transcript or genomic locus, respectively. On the other hand, levels of α-Syn could effectively be reduced by sequestering cytosolic monomers. The goal of an anti-prion is to deliver a non-toxic amyloid conformation, which competes with toxic amyloid species for the same pool of monomers [137,142]. Anti-prions offer advantages over the application of conventional therapeutics, which are inevitably metabolized after a certain period, whereas once applied, a benign, non-toxic amyloid conformation would self-template and accumulate in a prion-like fashion. As atomic resolution structural studies of pathogenic amyloid fibrils continue to emerge, our knowledge of non-toxic amyloid conformations is also likely to increase. If features of non-toxic amyloid strains were identified, one could use that knowledge to design improved anti-prions. Clinical cases in which patients displayed amyloid plaques distributed throughout the cerebral cortex in histopathological patterns akin to late-stage PD brains but lacked any neurological symptoms are known [143]. This potentially suggests the existence of non-toxic or less-toxic amyloid α-Syn species. Of course, this course of action would only be amenable for pathogenic, amyloidogenic proteins in which loss-of-function symptoms would be significantly less serious than the potential protection from a neurodegenerative disease, as indeed would potentially be the case for α-Syn [144].

A central application of amyloid strain identity in the context of human neurodegenerative diseases outside of direct clinical intervention could be diagnostic. As mentioned above, given the diversity and overlap of clinical presentation within the synucleinopathies and between other classes of neurodegenerative diseases (tauopathies, etc.), reaching an accurate early diagnosis is often difficult for clinicians; however, this would be an absolute necessity for any prospective therapeutic treatments (which do not currently exist) to be effective.

The search for disease surrogate biomarkers detectable in body fluids (blood or cerebrospinal fluid (CSF)) has been partially successful for some neurodegenerative diseases, especially in the case of AD. Here, levels of amyloid-β peptides (Aβ1-42), the presence of Tau phosphorylated at a certain residue (P-Tau181), or the total Tau presence in the CSF are routinely monitored in the clinical evaluation of AD [145]. For synucleinopathies, similarly reliable biomarkers of disease are currently not available; however, α-Syn strain identity could be utilized in this context in the future. Recent studies have exploited MSA- and PD-specific α-Syn amyloid conformations to develop a diagnostic test that can differentiate between either pathology with high specificity and sensitivity [113,146]. Small amounts of amyloid structures from patient CSF samples were amplified via PMCA followed by PK digest. The resulting band patterns, which were analyzed using Western blot technique, were highly consistent between MSA and PD patients respectively, but differed significantly between pathologies [113]. With the sensitivity of amyloid amplification-based methods improving and the understanding of disease-related strain identity increasing, amyloid strain “fingerprinting” may become a valuable diagnostic strategy not only for the synucleinopathies, but for other neurodegenerative diseases as well.

## 5. Summary and Outlook

Misfolding of the α-Syn protein into amyloid aggregates is intimately linked to the disease spectrum called the synucleinopathies. While the mechanisms of its misfolding, amyloid growth, and spreading throughout cells and tissues of the peripheral and central nervous system are slowly being uncovered, the symptomatic diversity and variability observed in synucleinopathy patients is still puzzling. The conformational strain hypothesis reviewed in this article may contribute to the explanation of that complexity seen in the clinical setting. Amyloidogenic proteins, from yeast to humans, seem to form a multitude of different conformations that assemble into distinct fibril variants or strains. Their individual structure can have considerable impacts on their biophysical, biochemical, and even biological properties, potentially dictating both pathology and clinical manifestations. In the context of α-Syn-related diseases, it is still not known how individual types of amyloids can target specific brain areas and cause a particular disease spectrum; however, there is increasing evidence that the cellular environment plays a critical role in determining the type of α-Syn variant that arises, and likewise how individual α-Syn variants affect specific cell types; therefore, it is important to further investigate the relationships between α-Syn conformers and their cellular environment [111]. In particular, the interactions of α-Syn strains with members of the cellular protein quality control pathway and molecular chaperones would be interesting to explore due to their dual role in amyloid aggregation [112]. Increased knowledge on the structural and pathophysiological properties of α-Syn assemblies will broaden our understanding of amyloid biology in general and holds great potential for the development of much needed diagnostic and therapeutic approaches in the future.

## Figures and Tables

**Figure 1 biomolecules-11-00931-f001:**
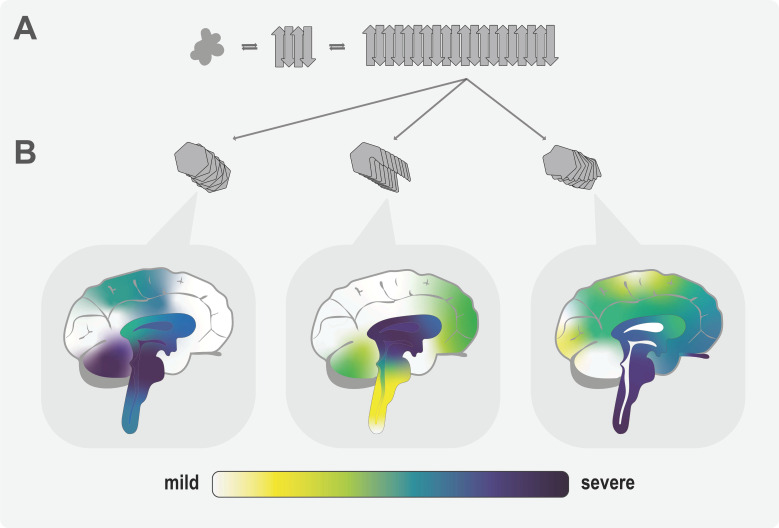
Graphical representation of the prion strain hypothesis: (**A**) a natively folded protein can acquire a β-sheet-rich conformation capable of templating misfolding of other native monomers; by incorporating additional monomers, oligomers form that further grow into amyloid fibrils; (**B**) the same native monomer can adopt various β-sheet-rich conformations; therefore, distinct three-dimensional fibril structures composed of differently folded subunits can arise. Following the prion strain hypothesis, these amyloid fibrils of different conformations affect different parts of the brain in characteristic ways. Proteinaceous inclusions and cellular damage would occur to varying degrees in different areas of the brain (yellow/mild → purple/severe).

**Figure 2 biomolecules-11-00931-f002:**
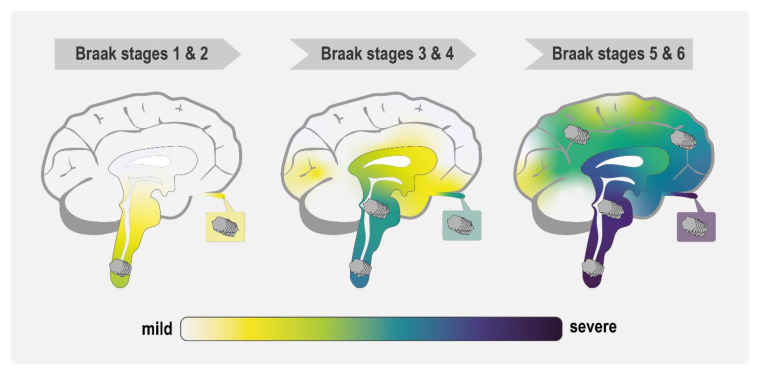
Schematic representation of α-Syn inclusion distribution in PD according to Braak staging. Brainstem and cortical Lewy bodies first appear in the lower brainstem and olfactory bulb of PD patients in Braak stages 1 and 2. In stages 3 and 4, Lewy bodies progressively accumulate in higher brainstem regions and in indicated cortex regions. Finally, PD patients in Braak stages 5 and 6 show accumulation of pathological α-Syn deposits throughout the cortex and brainstem. In the cartoon, the colored areas correspond to regions with α-Syn inclusions and neuronal loss, the intensity of which is indicated by the hue (yellow/mild → purple/severe).

**Figure 3 biomolecules-11-00931-f003:**
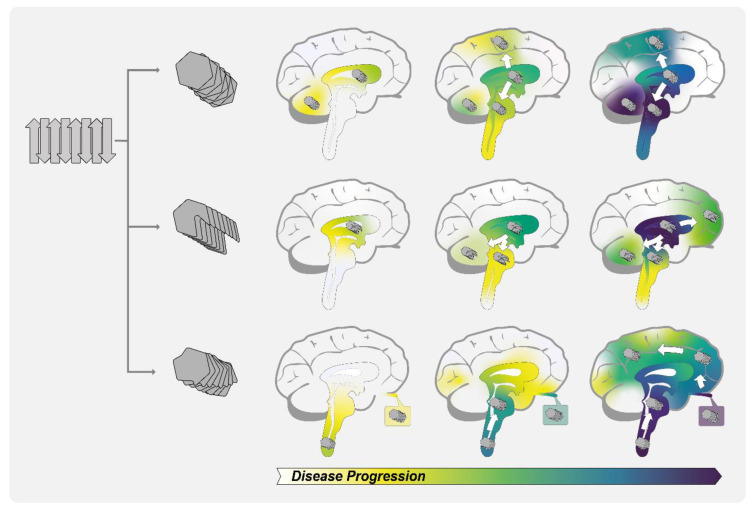
Graphical representation of α-Syn deposition patterns during the progression of various synucleinopathies based on conformation. Pathogenic α-Syn can adopt different conformations with different fibril architectures. According to the prion strain hypothesis, each synucleinopathy would be caused by a distinct α-Syn amyloid conformational variant or strain. Each strain has a distinct spreading pattern and associated neuropathology, resulting in different disease phenotypes in the respective patients. Synucleinopathy-specific α-Syn inclusions and cellular damage would accumulate to a characteristic extent in particular areas of the brain over the course of the disease (yellow/mild → purple/severe).

**Figure 4 biomolecules-11-00931-f004:**
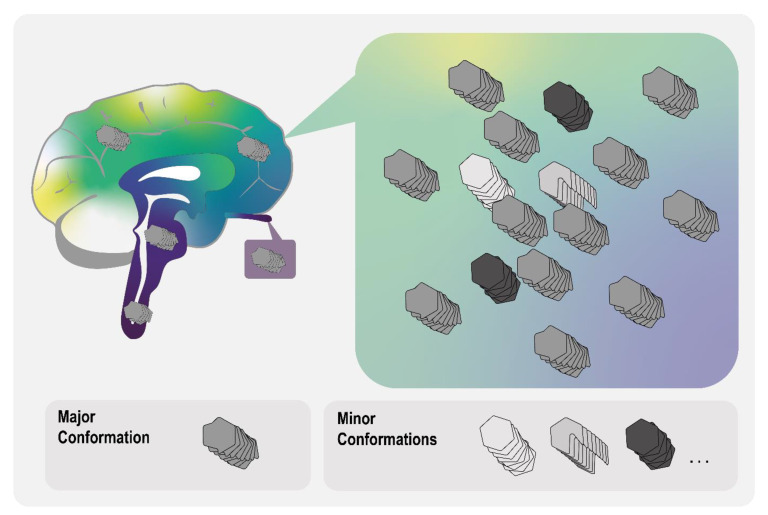
Graphical representation of the prion cloud model. Amyloid aggregates of the same protein may form a wide range of possible fibril or oligomer architectures. In prion and prion-like diseases, there likely exists one abundant major amyloid conformation, which would determine the strain characteristics, as well as a variety of less abundant minor conformations. In cases where a cellular degradation response is implemented against a certain major conformation, minor unaffected conformations may become more prevalent—a process referred to as strain mutation.

## Data Availability

Not applicable.

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
