# Peer review of "α-Synuclein Strains: Does Amyloid Conformation Explain the Heterogeneity of Synucleinopathies?"

_biomolecules, 2021, doi:10.3390/biom11070931_

Round 1

Reviewer 1 Report

This review by Hoppe and colleagues provides a thorough and very dense state-of-the-art of the research on the links between alpha-synuclein amyloids structure and the pathogenesis of synucleinopathies. It is well written and extremely (even a little too much) documented. This manuscript will for sure be useful for readers wanting a full bibliographic overview of this trendy field of research.

While I found in some instances the review too detailed (for example the entire description of yeast prions mechanisms), I raised below some lacks of critical discussion. I would have liked that the authors raise questions and enigmas still standing between amyloid proteins structures and human disorders phenotypes. These points and suggestions are the "major revisions" that I recommend to address. 

- In brief one mystery can be simplistically described as follows :

For prions : rarely fibrils (fibrillogenesis does not seem to represent a key pathogenic event in the case of PrP), immense panel of strain diversity, in human and lots of other species, tremendous lethality rate and short disease durations.

On the other side, alpha-synuclein : striking fibrillogenesis, many amyloid polymorphic structures, strickly human disorders, only 3 known synucleinopathies or “strains”, very low infectivity and no horizontal transmissibility.

Also, why are synucleinopathies restricted to humans ? A lack of aggressivity related to lifespan ? Why are they not “contagious” : then isn’t it PrP localization on the outer membrane of all neurons that makes it even its proper pathologic receptor, making prions so infectious…?

I personally think that the biohazard of experiments ran on prions is of course slowing down the research advances, but the points raised above are the true obstacles…

I have the feeling that the review would gain interest with clearly raising these questions.

- The part on yeast prions is interesting, but losing the reader in many unnecessary details… plus, the ambiguity of yeast prions is that they are pure fibrillar structures, while mammalian prions barely show amyloidogenesis ? In the end, the rare studies between prion aggregates’ structure and pathogenicity are based solely on the quaternary structure (Silveira 2005 etc), not on the conformation.

- Authors should comment on the fact that “ribbons” are not physiologically coherent. The parallels between ribbons and synucleinopathy amyloids is trendy but has to be put in perspective of the impossibility of forming and replicating exactly these structures in vivo…

- Also I am surprised that some extremely important recent articles in the field are not cited, and would highly recommend to include in the discussions the following papers, that contribute in my opinion to the synuclein strain-structure-function field :

>Lau et al, PMID 31792467. About strain structural, biochemical, and pathological properties, and their conservation on passages.

>Sokratian et al, PMID 33641009. Structural analysis of how seeding and fibril-templating activity relates to disease progression rate.

>De Giorgi et al, PMID 33008896. Links structure-bioactivity of fibrils, and replication of polymorphs. Important ThT-negative fibrils (in relation with MSA ThT- in Shahnawaz et al).

- In their figures, authors show always (in a single brain and over time and space) a mono-structural amyloid existence. In the case of prions the hypothesis is that a pool of conformations co-exist and are selected by cell types / pathways / host protein conformational compatibility (as explained line 525). Could authors illustrate the possible existence of multiple conformations in one brain / area / cell / or over time ?

- One title typo:

On titles 3., 3.1, and 4. is written “α-. Syn” probably auto-corrected from “α-Syn”.

Author Response

Reviewer #1

This review by Hoppe and colleagues provides a thorough and very dense state-of-the-art of the research on the links between alpha-synuclein amyloids structure and the pathogenesis of synucleinopathies. It is well written and extremely (even a little too much) documented. This manuscript will for sure be useful for readers wanting a full bibliographic overview of this trendy field of research.

While I found in some instances the review too detailed (for example the entire description of yeast prions mechanisms), I raised below some lacks of critical discussion. I would have liked that the authors raise questions and enigmas still standing between amyloid proteins structures and human disorders phenotypes. These points and suggestions are the "major revisions" that I recommend to address. 

- In brief one mystery can be simplistically described as follows :

For prions : rarely fibrils (fibrillogenesis does not seem to represent a key pathogenic event in the case of PrP), immense panel of strain diversity, in human and lots of other species, tremendous lethality rate and short disease durations.

On the other side, alpha-synuclein : striking fibrillogenesis, many amyloid polymorphic structures, strickly human disorders, only 3 known synucleinopathies or “strains”, very low infectivity and no horizontal transmissibility.

Also, why are synucleinopathies restricted to humans? A lack of aggressivity related to lifespan ? Why are they not “contagious” : then isn’t it PrP localization on the outer membrane of all neurons that makes it even its proper pathologic receptor, making prions so infectious…?

I personally think that the biohazard of experiments ran on prions is of course slowing down the research advances, but the points raised above are the true obstacles…

I have the feeling that the review would gain interest with clearly raising these questions.

We thank the reviewer for the inspiration and asking these important questions. We have addressed these aspects in the following paragraph, which we believe will increase interest in our review: “As fundamental aspects of prion biology seem to be shared both by PrP and α-Syn, the question arises why the associated diseases, TSEs and synucleinopathies, have such drastically different clinical manifestations. Why are TSEs so much more aggressive, usually leading to akinetic mutism and subsequent death within a year of the onset of symptoms, while synucleinopathies have a relatively slow progression? And why are only prion diseases infectious? Biochemical and high resolution imaging techniques agree that both pathological PrPSc and α-Syn material contain amyloid fibrils. However, their individual size, growth kinetics, frangibility and environmental stability could differ, affecting their propagation capacity and ultimately determining disease dynamics and infectivity [133]. Greater accessibility of a given fibril structure for the cellular quality control system (e.g., chaperone networks) could lead to higher fibril fragmentation [134,135]. This could explain the pathological differences between the two disease classes, as small aggregates or oligomers in related amyloid proteins have been associated with accelerated spreading and higher toxicity [127]. The individual fibril architectures could further influence the interaction with particular essential proteins and contribute to different degrees of toxicity in distinct brain areas. In addition, other factors such as cellular localization (at the outer cell membrane in the case of PrPCvs. cytosol in the case of α-Syn), expression levels, and cellular functions are also likely to play a role.”

- The part on yeast prions is interesting, but losing the reader in many unnecessary details… plus, the ambiguity of yeast prions is that they are pure fibrillar structures, while mammalian prions barely show amyloidogenesis ? In the end, the rare studies between prion aggregates’ structure and pathogenicity are based solely on the quaternary structure (Silveira 2005 etc), not on the conformation.

We have shortened this part by removing the explanation about the red/white colony color assay.

- Authors should comment on the fact that “ribbons” are not physiologically coherent. The parallels between ribbons and synucleinopathy amyloids is trendy but has to be put in perspective of the impossibility of forming and replicating exactly these structures in vivo…

We agree with the reviewer that in vitrogenerated fibrils may not form physiologically relevant structures. Therefore, we already stated in the original version of the manuscript “However, it is unclear whether the fibrils generated in a test tube fully represent the structures of amyloid α-Syn in cellular deposits in patients and whether they are indeed structurally distinct in different synucleinopathies.” We have now added the following sentence to particularly highlight the issue with ribbons: “For example, “ribbons” are generated under non-physiological salt concentrations, making it questionable whether this structure can be formed and replicated in vivo[46].”

- Also I am surprised that some extremely important recent articles in the field are not cited, and would highly recommend to include in the discussions the following papers, that contribute in my opinion to the synuclein strain-structure-function field :

>Lau et al, PMID 31792467. About strain structural, biochemical, and pathological properties, and their conservation on passages.

>Sokratian et al, PMID 33641009. Structural analysis of how seeding and fibril-templating activity relates to disease progression rate.

>De Giorgi et al, PMID 33008896. Links structure-bioactivity of fibrils, and replication of polymorphs. Important ThT-negative fibrils (in relation with MSA ThT- in Shahnawaz et al).

We thank the reviewer for the suggestions. We have added the recommended article citations as follows:

- Lau et al.: “Indeed, subsequent studies revealed that these in vitrogenerated α-Syn polymorphs were able to affect different brain regions and induce distinct disease phenotypes that are conserved over serial passaging in rodents [47,48,102].”

- Sokratian et al.: “Differences in seeding capacity do not only exist between individual synucleinophathies. High heterogeneity in fibril templating activity has also been reported in CSF and brain extracts exclusively from DLB patients, which may be a contributing factor to the variable disease phenotypes [107]. A high fibril seeding activity correlated with early mortality [108].”

- De Giorgi et al.: “Another important point regarding in vitro amplification methods is that the standard procedure involves the use of thioflavin T (ThT) for for fibril detection. However, α-Syn is capable of forming ThT-invisible polymorphs that escape ThT based monitoring, which could lead to misinterpretations regarding the structure and amount of fibrillar α-Syn in patient-derived samples [113,118].”

- In their figures, authors show always (in a single brain and over time and space) a mono-structural amyloid existence. In the case of prions the hypothesis is that a pool of conformations co-exist and are selected by cell types / pathways / host protein conformational compatibility (as explained line 525). Could authors illustrate the possible existence of multiple conformations in one brain / area / cell / or over time ?

To make the part about the prion cloud hypothesis more intuitive, we have added a fourth figure illustrating the basic concept and showing the existence of multiple conformations. The figure also shows that there is one major and several minor conformations within a brain area. However, these are only visible when zoomed in, whereas in the overview only the main conformation is visible. We do also speculate about a potential existence of multiple conformations of α-Syn in synucleinopathies as suggested by reviewer 4: “Since amyloidogenic proteins share many fundamental features (discussed above), it is tempting to extend this hypothesis beyond bona fide prions [141] to α-Syn strains in synucleinopathies, although there is no experimental evidence for this to date”.

- One title typo:

On titles 3., 3.1, and 4. is written “α-. Syn” probably auto-corrected from “α-Syn”.

We have corrected the typo.

Reviewer 2 Report

The authors of the review entitled “α-synuclein conformational strains: does amyloid structure define the pathology of synucleinopathies?” provide a comprehensive overview on the prion strain hypothesis in the context of alpha synuclein, with the main focus on a possibility of causing different synucleinopathies by distinct alpha synuclein strains/conformers. The authors start introducing the reader to the basic background regarding alpha synuclein physiological function, its pathology and clinical heterogeneity. They finely explain a definition of a prion strain, templating phenomenon and its origin going back to early findings about Prion Protein (PrP) in mammals and Sup35 in yeast. The authors do an extensive review on a current body of evidence suggesting that alpha synuclein exists in in the forms of several strains and acts in a prion-like manner in vitro and in vivo. They speculate about a possibility of synucleinopathy transmission within and between species based on the knowledge of incidents of Prion Protein infections. In the last part of the manuscript the authors talk about potential implications of the strain knowledge in clinical diagnostics. The manuscript provides an exhaustive summary of the up to date knowledge about alpha synuclein in the context of a prion hypothesis. Besides a few minor comments, this manuscript will be a useful review for many in the field and is suitable for publishing.

Major points:

  1. The title of the manuscript should be reworked. Based on the title the reader expects more insights about current structures of alpha synuclein conformers. The authors mention only a structure of MSA fibrils but without reviewing other known so far. The manuscript focuses on a prion hypothesis instead (not structures), then it should be the main clue in the title.

Minor points:

  1. In the part “Introduction”, line 36, the authors recall known missense mutations in SNCA gene which underlie inherited early-onset PD (and others later on). The mutation “G51N” is misspelled, since there is only G51D known causing the disease. Following that, it would be suitable to mention about all known alpha synuclein mutations and fulfill the gap about: A18T, A29S, A53V and E83Q which are the most recent findings and have not been mentioned.
  2. In the lines 78-83 the authors talk very briefly about clinical symptoms of MSA, next to more insightful discussion about PD and LBD. It would be expected to discuss two major types of MSA which are seen in clinics, which are parkinsonian type (MSA-P) and cerebellar type (MSA-C).
  3. In the part “What is a prion?”, the “prion” itself should be explained, that it comes from a word fusion of “protein” and “infection”.
  4. In the lines 246, 247 and 508, “α-. Syn” should be corrected to “α-Syn”.
  5. The extra density in the MSA structure (which is one of the characteristics differ from the recombinant fibrils) should be mentioned earlier. It should be included in the section on lines 363-379 where the authors discuss the structure.
  6. In the part “Can synucleinopathies be infectious?” it is desired to discuss current knowledge about gut-brain connection through a vagus nerve and a possible risk of eating poultry and beef since those animals in the wild type sequence has threonine in the position 53 instead of alanine in humans which may increase seeding potency.

Author Response

Reviewer #2

The authors of the review entitled “α-synuclein conformational strains: does amyloid structure define the pathology of synucleinopathies?” provide a comprehensive overview on the prion strain hypothesis in the context of alpha synuclein, with the main focus on a possibility of causing different synucleinopathies by distinct alpha synuclein strains/conformers. The authors start introducing the reader to the basic background regarding alpha synuclein physiological function, its pathology and clinical heterogeneity. They finely explain a definition of a prion strain, templating phenomenon and its origin going back to early findings about Prion Protein (PrP) in mammals and Sup35 in yeast. The authors do an extensive review on a current body of evidence suggesting that alpha synuclein exists in in the forms of several strains and acts in a prion-like manner in vitro and in vivo. They speculate about a possibility of synucleinopathy transmission within and between species based on the knowledge of incidents of Prion Protein infections. In the last part of the manuscript the authors talk about potential implications of the strain knowledge in clinical diagnostics. The manuscript provides an exhaustive summary of the up to date knowledge about alpha synuclein in the context of a prion hypothesis. Besides a few minor comments, this manuscript will be a useful review for many in the field and is suitable for publishing.

Major points:

  1. The title of the manuscript should be reworked. Based on the title the reader expects more insights about current structures of alpha synuclein conformers. The authors mention only a structure of MSA fibrils but without reviewing other known so far. The manuscript focuses on a prion hypothesis instead (not structures), then it should be the main clue in the title.

We thank the reviewer for the constructive feedback. We have reworked the title to avoid the word structure: “α-Synuclein Strains: Does Amyloid Conformation Explain the Heterogeneity of Synucleinopathies”. Together with the abstract, it should now be very clear what the review is about.

Minor points:

  1. In the part “Introduction”, line 36, the authors recall known missense mutations in SNCA gene which underlie inherited early-onset PD (and others later on). The mutation “G51N” is misspelled, since there is only G51D known causing the disease. Following that, it would be suitable to mention about all known alpha synuclein mutations and fulfill the gap about: A18T, A29S, A53V and E83Q which are the most recent findings and have not been mentioned.

We have corrected the typo and added the recent findings according to the reviewer’s suggestions. The respective part of the introduction now states: “The causal role of α-Syn in disease pathogenesis is supported by genetic evidence. Several missense mutations in the human SNCA gene that lead to destabilizing single amino acid changes in the protein (A53T, A30P, E46K, H50Q, G51D and A53V) are linked to autosomal-dominantly inherited early-onset PD [8–13]. Furthermore, A18T and A29S amino acid substitutions have been recently described in late onset sporadic PD patients [14]. In addition, multiplications (duplication or triplication) of the SNCA gene locus have also been associated with early-onset familial parkinsonism [13,15–18]. An elevated SNCA gene dosage correlates with increased disease penetrance, earlier disease onset and more severe PD symptoms [19,20]. Additional evidence, albeit less prominent, points to involvement of α-Syn levels and mutations in the etiology of DLB and MSA. For instance, A53T, E46K, and E83Q missense mutations and SNCA multiplications have been observed in families with mixed PD and DLB phenotypes [21,22], while G51D and A53E mutations have been found in cases with mixed PD/MSA pathogenesis [23,24]”.

  1. In the lines 78-83 the authors talk very briefly about clinical symptoms of MSA, next to more insightful discussion about PD and LBD. It would be expected to discuss two major types of MSA which are seen in clinics, which are parkinsonian type (MSA-P) and cerebellar type (MSA-C).

We have added the following description of MSA-P and MSA-C: “MSA is further divided into two subtypes depending on the type of symptoms present at the time of diagnosis: the parkinsonian subtype (MSA-P) that is characterized by parkinsonism and the cerebellar subtype (MSA-C), where cerebellar ataxia is prominent [44].”

  1. In the part “What is a prion?”, the “prion” itself should be explained, that it comes from a word fusion of “protein” and “infection”.

We have added an explanation of the term “prion” to the part “What is a prion?”: “However, as the word prion (derived from “proteinaceous infectious particle”) already implies, a unique feature of prion diseases is that they can be transmitted within a species (< 1% of cases) and even between species in certain circumstances.”

  1. In the lines 246, 247 and 508, “α-. Syn” should be corrected to “α-Syn”.

We have corrected the typo.

  1. The extra density in the MSA structure (which is one of the characteristics differ from the recombinant fibrils) should be mentioned earlier. It should be included in the section on lines 363-379 where the authors discuss the structure.

We are now mentioning the extra density in the earlier section: “Intriguingly, the protofilaments form a cavity encompassing an additional density of unknown origin, which indicates the presence of a non-proteinaceous, yet unidentified molecule [116]”. We have therefore also changed the sentence in the next paragraph: “The presence of additional cellular components could have an influence on the amyloid conformation obtained in vivo, as indicated by the aforementioned structural data of extracted MSA filaments analyzed directly without an in vitroamplification step [116,117].”

  1. In the part “Can synucleinopathies be infectious?” it is desired to discuss current knowledge about gut-brain connection through a vagus nerve and a possible risk of eating poultry and beef since those animals in the wild type sequence has threonine in the position 53 instead of alanine in humans which may increase seeding potency.

Because there is no epidemiologic evidence that consumption of poultry and beef leads to a higher incidence of synucleinopathies, we respectfully prefer not to discuss this so as not to alarm the reader.

Reviewer 3 Report

Lines 115-119. The amplification of the aggregation from the PrPSc templates has been modelled but not experimentally demonstrated. It should be clearly stated here. 

Author Response

Reviewer #3:

Lines 115-119. The amplification of the aggregation from the PrPSc templates has been modelled but not experimentally demonstrated. It should be clearly stated here.

We thank the reviewer for pointing out this inaccuracy to us. The sentence was qualified as follows: “In this way, PrPScamyloid aggregates are thought to grow on either end by monomer addition [54,55].

Reviewer 4 Report

This review by Dr. Nussbaum-Kramer and colleagues summarizes current knowledge on the prion strain concept in alpha-synucleinopathies. The manuscript is very well written, perfectly summarizes the current state of the art in the field, and will certainly prove to be valuable to a wide range of readers interested in prions, proteinopathies or neurodegenerative diseases.

I only have two minor comments:

  • The authors state that alpha-synuclein is the main/major component of pathological inclusions such a Lewy bodies (e.g. abstract, line 63). This may not be exactly true as recent cryo-electron tomograpy data (e.g. Trinkaus et al., 2021) show that these inclusions contain a lot of other constituents including proteins and lipid vesicles.
  • The authors mention the 'cloud' hypothesis of coexisting prion conformers in single individuals. The authors may discuss whether this hypothesis could be extended (or not?) to alpha-synuclein with multiple alpha-synuclein strains coexisting in an individuals brains and explaining different propagation and neuropathological patterns.

Author Response

Reviewer #4

This review by Dr. Nussbaum-Kramer and colleagues summarizes current knowledge on the prion strain concept in alpha-synucleinopathies. The manuscript is very well written, perfectly summarizes the current state of the art in the field, and will certainly prove to be valuable to a wide range of readers interested in prions, proteinopathies or neurodegenerative diseases.

I only have two minor comments:

  • The authors state that alpha-synuclein is the main/major component of pathological inclusions such a Lewy bodies (e.g. abstract, line 63). This may not be exactly true as recent cryo-electron tomograpy data (e.g. Trinkaus et al., 2021) show that these inclusions contain a lot of other constituents including proteins and lipid vesicles.

We thank the reviewer for the very positive feedback. We are aware that inclusions found in synucleinopathies contain other proteins and lipid vesicles, but did not mention this explicitly, as we should have. The abstract now reads “Synucleinopathies are a heterogeneous group of neurodegenerative diseases with amyloid deposits that contain α-synuclein (SNCA/α-Syn) protein as a common hallmark”. We have also changed the respective sentence in the introduction and added two more citations (Shahmoradian et al., 2019 and Trinkaus et al., 2021): “Although α-Syn is a major protein component of pathological inclusions along with membranous organelles [36,37], they differ not only in their subcellular localization but also in the brain regions affected in the individual diseases.”

  • The authors mention the 'cloud' hypothesis of coexisting prion conformers in single individuals. The authors may discuss whether this hypothesis could be extended (or not?) to alpha-synuclein with multiple alpha-synuclein strains coexisting in an individuals brains and explaining different propagation and neuropathological patterns.

As suggested by the reviewer, we now speculate about a potential existence of multiple conformations of α-Syn in synucleinopathies: “Since amyloidogenic proteins share many fundamental features (discussed above), it is tempting to extend this hypothesis beyond bona fide prions [141] to α-Syn strains in synucleinopathies, although there is no experimental evidence for this to date”.

Round 2

Reviewer 1 Report

Authors addressed all my comments, the manuscript is very thorough and documented, but remains easy to read.

I appreciate the new figure provided for explaining the structural landscape of amyloids.